# Peripartum Predictors of the Risk of Postpartum Depressive Disorder: Results of a Case-Control Study

**DOI:** 10.3390/ijerph17238726

**Published:** 2020-11-24

**Authors:** Kornelia Zaręba, Jolanta Banasiewicz, Hanna Rozenek, Stanisław Wójtowicz, Grzegorz Jakiel

**Affiliations:** 1First Department of Obstetrics and Gynecology, Center of Postgraduate Medical Education, 01-813 Warsaw, Poland; grzegorz.jakiel1@o2.pl; 2Department of Medical Psychology and Medical Communication, Medical University of Warsaw, 02-091 Warsaw, Poland; jolantabanasiewicz.wum@gmail.com (J.B.); hanna.rozenek@wum.edu.pl (H.R.); stwojt@o2.pl (S.W.)

**Keywords:** postpartum depression, baby blues, postpartum mood disorders, predictors of postpartum depression, risk factors of postpartum depression, EPDS

## Abstract

Background: The study aimed at the identification of the risk factors present during delivery, which might be present in prophylactic programs concerning postpartum mood disorders. Material and Method: This was a retrospective comparative study. The study material included data retrieved from the medical records of patients hospitalized in the Teaching Department of Gynecology and Obstetrics of Professor Orłowski Hospital in Warsaw, in the years 2010–2017. The EPDS data of 604 patients were analyzed. The study group included 75 women who obtained at least 12 points in the EPDS and the control group was made up of 75 women who obtained no more than 5 points in the EPDS. Results: The women in whom we noted an increased risk of developing mood disorders had blood loss >1000 mL and had a significantly longer stage II and III of labor than the control group. Other risk factors were cesarean section, vaginal delivery with the curettage of the uterine cavity, slightly lower APGAR scores (0.4 pts), and lower birth weight (approximately 350 g) of the child. Women at a low risk of postpartum mood disorders more commonly underwent episiotomy during delivery (76%). Conclusions: Increased supervision and support should be offered to women who experienced the above-mentioned risk factors.

## 1. Introduction

Postpartum mood disorders occur in a large group of women and constitute the most common emotional disorder developing after delivery [1]. Depressive (10–40%) and psychotic (0.1–0.2%) disorders are much less common. According to numerous clinicians, postpartum mood worsening might increase the risk of developing postpartum depression and subsequent anxiety disorders [2]. The symptoms of postpartum depression developed in about 20% of women, with the diagnosis of postpartum mood worsening [3,4]. Postpartum mood depression was named “a physiological side effect of a woman developing into a mother” by some authors [5].

Some researchers view it as a physiological variant resulting from hormonal changes [3]. They are associated with changes within the hypothalamic–pituitary–adrenal axis, particularly with the increased production of placental CRH (corticotropin releasing hormone), increased secretion of ACTH (adrenocorticotropic hormone) during pregnancy, and the resultant decreased estriol, progesterone, and CRH for up to 6 days postpartum, and persistent high levels of cortisol [6]. ACTH decreases for up to 3 days after delivery, and then it increases again. A clinical trial showed a negative correlation between the occurrence of postpartum depressive disorders (PPD) and the level of estriol, and a positive correlation with the ACTH level. The concentrations of estradiol, progesterone, and free estriol, decreased by approximately 90–95% over a few days after delivery [7]. Other risk factors of postpartum depressive disorders include biological ones related to the pregnancy and delivery, and environmental factors.

The biological factors of baby blues include pain resulting from perineal or abdominal wound healing, shrinking of the uterus, and pain of the breasts and the spine, associated with the change in body statics in women after delivery. Its etiology might also be due to the elevated levels of monoamine oxidases and reduced serotonergic activity, immediately after delivery [8]. Other important factors that play an important role are linked to the health status, care, behavior, and feeding of the neonate. The course of lactation in women in labor is important in this case—lactation onset, the production of breast milk that is sufficient for the neonate, the perception of one’s own breastfeeding effectiveness, and related difficulties and pain. No correlation was found between this type of disorders and the method of feeding the neonate. The onset of postpartum mood worsening is frequently associated with the occurrence of difficulties with regards to breastfeeding or the overproduction of breast milk [9].

Significant psychological factors include mental predisposition, especially towards anxiety and depression, fear for the health and life of the child, and the feeling of uncertainty related to changes occurring in life. The feeling of reduced attractiveness and the increased body weight compared to pregestational weight is a significant risk factor [9]. A positive history for depression, PMS (premenstrual syndrome), and mood disorders related to pregnancy were also confirmed to be the risk factors [9]. Numerous factors contribute to the development of postpartum mood worsening. PPD is a response to the experienced physical stress—rapid hormonal changes, pain, and fatigue; and the psychological stress associated with the new role in life. This might result from the reduced perception of one’s attractiveness connected with the changes in the body observed by the woman in labor. Anxiety could also be related to the change of previous lifestyle and the associated feeling of insufficient competences necessary to perform the role of the mother. The intensification of PPD manifestations is commonly coexistent with fatigue related to the fact that the woman performs everyday care of the neonate and returns to her daily responsibilities at the same time. Some authors claimed that the development of postpartum mood worsening in women at this stage of life was associated with the discomfort resulting from the feeling of the lack of freedom and independence [5].

Important roles in the etiology of postpartum mood disorders are also played by environmental factors, particularly including the support provided to the woman in labor by the medical personnel and the family and friends (the partner, the mother). Other factors that increase the risk are—young maternal age, the lack of social support, financial problems, stressful situations during pregnancy, marital problems, domestic violence, and addictions [10]. A significant role is also played by emotional support, financial support, intelligence support, and empathic relations [11].

Postpartum depressive disorders are characterized by a mild course and are limited to the first weeks after delivery. The first symptoms usually appear about 3–4 days after delivery, and persist for up to several weeks. The most common symptoms include—mood worsening, tearfulness, emotional lability, irritability, increased susceptibility to frustration, headaches, concentration problems, anorexia, and sleep disturbances [9,12].

Numerous pediatricians emphasized the significance of the first 1000 days of the child’s life for its further emotional development [13]. The mental status and frame of mind of a woman after delivery are highly significant for the entirety of the physical and mental health of the child and the functioning of the whole family.

## 2. Materials and Methods

### 2.1. The Aim of the Study

The present study is a part of the research of the perinatal and postnatal determinants of developing postpartum mood worsening. All subjects gave their informed consent for inclusion before they participated in the study. The study was conducted in accordance with the Declaration of Helsinki, and the protocol was approved by the Ethics Committee of the Center of Postgraduate Medical Education in Warsaw (Research Number 63/PB/2017). The study aimed at the identification of the predictors of mood worsening in women during the first week postpartum. Therefore, the following research questions were proposed:What parameters of the neonatal status are correlated with the woman’s mood during the first postpartum week?Is the mode of delivery correlated with the woman’s mood during the first postpartum week?Are obstetric complications, such as episiotomy and the curettage of the uterine cavity, correlated with the woman’s mood during the first postpartum week?Is the length of hospital stay after delivery correlated with the woman’s mood during the first postpartum week?

### 2.2. Material

Study material included data retrieved from medical records of patients hospitalized in the Teaching Department of Gynecology and Obstetrics of Professor Orłowski Hospital in Warsaw in the years 2010–2017. The medical records were analyzed in terms of selected data, i.e., demographic variables, variables concerning the course of delivery and the early postpartum period, and variables concerning the neonatal status. The EPDS scale is routinely attached to the medical records of the department. Women in labor who are hospitalized in the Department are asked to complete the questionnaire as part of a screening testing, 2–4 days after delivery. The aim of asking patients to complete the EPDS was to obtain information concerning the current frame of mind of postpartum patients and, if necessary, to provide a suitable form of psychological, or psychiatric support. Only patients from whom informed consent could be obtained were qualified for the EPDS test. The exclusion criterion for the EPDS test was no knowledge of Polish, which prevented questionnaire completion.

We analyzed data included in the medical records of 604 patients who expressed their consent for the use of their personal data in research. The study group included 75 women who obtained at least 12 points in EPDS (mean = 14.92, standard deviation = 3.05). The control group was made up of 75 women who obtained no more than 5 points in the EPDS. Along with the score, we analyzed the age, marital status, and the place of residence.

### 2.3. Research Tool Description

The Edinburgh Postnatal Depression Scale (EPDS) is a tool that is most commonly used in the assessment of the risk of developing postpartum depressive disorders (Appendix A), which is also recommended by the American College of Obstetricians and Gynecologists (2015). The scale was authored by Cox et al. [14]. The assessment of patients hospitalized in the Department of Obstetrics and Gynecology of Professor Orłowski Independent Public Teaching Hospital in Warsaw was performed with the use of the Polish version of the scale, translated by Maria Bnińska, MSc., PhD [15]. The scale was used for the assessment of the frame of mind over the last week, prior to the test. The EPDS is a short scale. It comprises 10 statements describing various aspects of the frame of mind of women, such as anhedonia, sense of guilt, anxiety, panic attacks, exhaustion/overtiredness, sleep disturbance, sadness/dejection, tearfulness, and suicidal thoughts. Each answer is scored from 0 to 3 points. The total of all obtained points is the general score (maximum 30 points). The higher the score, the higher the risk of developing postpartum depression in the tested person. The borderline value indicating an increased risk of postpartum depressive disorders was assumed to be 12 points, with the reservation that some specialists recommended vigilance even in case of scores lower by several points [16,17]. Moreover, it was emphasized that particular attention is necessary in cases when the woman reported the presence of suicidal ideation, even if the total EPDS was low [16]. The tool was characterized by good psychometric properties. According to the original research, the sensitivity was 86%, the specificity was 78%, and the Cronbach’s alpha coefficient was 0.88. The authors of the EPDS and the British Journal of Psychiatry, who were the owners of copyright, accepted the use and copying of the tool, provided that the source was cited [16].

### 2.4. Statistical Analysis of Data

Pairwise selection method was used with the criterion variables were age, level of education, and marital status. No tested variables (apart from the duration of the first stage of delivery) presented normal distribution (Kolmogorov–Smirnov test). Therefore, comparative analyses between the groups were performed with the use of non-parametric tests—the Mann-Whitney U test for variables measured with ratio scales and the chi-square test for nominal variables. The study was conducted with the use of IBM SPSS Statistics for Windows, Version 24.0., Armonk, NY, USA: IBM Corporation (Released 2016).

## 3. Results

### 3.1. Demographic Data

The average age of the participants was 30.6 years (SD = 4.70). The majority of patients completed tertiary education (66%). Most of them were married (83%) and over half of them lived in a city (54.7%) (Table 1). Pregnancy duration was 27–50 weeks (mean = 38.56; SD = 2.50).

### 3.2. Peripartum Risk Factors

The women in whom we noted an increased risk of developing mood disorders were hospitalized significantly longer (*p* < 0.002), with the difference being almost two days. Regrettably, the time of hospitalization was a variable analyzed after measuring EPDS. Therefore, it could not be determined as a risk factor. Study group participants lost considerably more blood during delivery (over 1000 mL) than the control group patients. Blood loss was also a factor predisposing to mood worsening (Table 2). Women who were found to be at an increased risk of developing postpartum depressive disorders, lost an average of 400 mL of blood during delivery. Women whose EPDS score indicated a low probability of developing postpartum depressive disorders, lost slightly less than 300 mL of blood during delivery. No significant correlation was noted between the duration of the first stage of delivery and the development of postpartum depressive disorders. It lasted longer than in the control group. Two remaining stages of delivery lasted longer in the group of women, with an increased risk of developing postpartum mood disorders. The differences were statistically significant (*p* < 0.07) in case of the second stage of delivery (*p* < 0.07) and the third stage of delivery (*p* < 0.07) (Table 2).

Almost half (46.7%) of the participants who were at an increased risk of developing postpartum mood disorders underwent Cesarean section. With regards to the control group, operative delivery was noted only in the case of 12% of participants. The difference was statistically significant (*p* < 0.0001) (Table 3).

Women at a low risk of postpartum mood disorders underwent episiotomy during delivery, more commonly (76%) than women who were at an increased risk of developing postpartum mood disorders (47%). The difference was statistically significant (*p* < 0.0001) (Table 4).

The group of women who were at an increased risk of postpartum depressive disorders significantly more commonly (37.9% vs. 17%), underwent an instrumental revision of the uterine cavity after delivery (*p* < 0.04) (Table 5).

### 3.3. Neonatal Risk Factors

The offspring of women who were at an increased risk of postpartum mood disorders had slightly lower APGAR scores (0.4 pts) and a lower birth weight (approximately 350 g) than the offspring of women who were not at an increased risk of postpartum mood disorders. The differences were statistically significant (*p* < 0.04 and *p* < 0.008, respectively) (Table 6).

We also carried out stepwise logistic regression. With regards to the analyzed variables, neonatal weight was the only significant (although not very strong) factor, which underlay high or low EPDS scores.

## 4. Discussion

Research conducted so far is equivocal with regards to the results concerning the risk factors of postpartum depressive disorders.

The present study demonstrated that an increased risk of postpartum depressive disorders occurred in women who had undergone Cesarean section. Similar results were obtained by researchers in Thailand and India, where postpartum mood disorders were reported in 8.23% of women in the vaginal delivery group and in over 21% of women in the Cesarean section group [18]. Similarly, Malik et al., reported the risk of postpartum depression assessed with EPDS to be significantly higher in women after Cesarean section than in the case of vaginal delivery [19]. A study conducted in Romania in women with the diagnosis of postpartum depression, demonstrated that the percentage was slightly higher after Cesarean section (51.3%) than after vaginal delivery (46%). However, it was not a statistically significant difference [19]. A study conducted in rural India showed that women who had had a vaginal delivery reported a higher postpartum quality of life (QoL), compared to women who had undergone Cesarean sections [20]. Numerous studies showed that the quality of life, especially with regards to physical health, negatively correlated with the frequency of postpartum depression [21]. However, Wiklund et al. conducted a study using the EPDS in 558 women in labor, and observed no significant differences in the occurrence of postpartum depression between women who had had vaginal delivery or Cesarean section. Similarly, Chen et al. reported no increased risk of postpartum depression associated with the mode of delivery or neonatal status in patients with pre-eclampsia, during pregnancy. However, the researchers noted a 3- to 4-fold increase of the risk associated with the primary disease [22].

A study by Sun et al., also showed no influence of episiotomy on the increased prevalence of PPD [23]. In Poland, episiotomy was performed during about 57% deliveries [24]. The present study showed that episiotomy was a factor that reduced the risk of postpartum mood disorders. Such a result was not reported in any available papers. It might be due to the shorter second stage of delivery. Additionally, such a result might be a result of preparing women for the possibility of episiotomy, which might result in a positive attitude towards the procedure and the feeling of being in control. One of the aims of episiotomy is to protect the perineum from rupture. The role is fulfilled when the procedure is performed correctly, at a suitable moment, and with appropriate care afterwards. Those factors might contribute to the postpartum situation of study participants, with regards to the exacerbation of physical symptoms, which are usually experienced by women in this period. The majority of researchers who studied this issue indicated the opposite correlation, which pointed at episiotomy as a risk factor of developing postpartum depression. Alomar stated that pain associated with the healing of episiotomy wound might contribute to the development of depressive disorders [25]. According to the literature, mood worsening is also attributed to anger with the medical personnel for not preventing episiotomy, which is viewed in a negative way. Episiotomy might trigger grief related to the fact that the woman was unable to deliver without such a medical intervention [25]. Moreover, Signorello indicated better psychological and sexual functioning in women whose perineum was protected during delivery.

Sun et al., conducted a study in China, which demonstrated that a significantly lower prevalence of postpartum depression was noted in patients who were administered epidural anesthesia during delivery [23]. The study also showed that those women also attempted to breastfeed earlier. The authors explained that lower stress associated with the anesthesia led to smaller catecholamine release, which, in turn, translated into a more rapid prolactin increase. Obviously, anesthesia prolonged the delivery, but it had no effect on the possibility of intensifying postpartum depressive disorders. A study conducted by Boudou et al. indicated the presence of a correlation between labor pains and the prevalence of baby blues. However, the correlation was more significant for the affective nature of pain (r = 0.32, *p* < 0.05) than its sensory character (r = 0.28, *p* < 0.05) [26]. The authors suggested that patients might view childbirth pain as a failure, as they expected their delivery to be painless. The authors also stated that the intensity of postpartum baby blues was the best predictor of postpartum depression. The present study showed no significance of this kind.

However, a significant correlation was found between the amount of blood lost during delivery and an increased risk of developing postpartum depression. The duration of labor and the amount of blood lost translated into the somatic condition of the patient (physical capacity). Additionally, it might prolong hospitalization, which significantly affects the mental status. In other words, longer labor, higher blood loss, and longer hospitalization, increased the probability of deteriorating the mental health status of a woman in labor. A study by Mohammad conducted in 353 women in Jordan also showed a correlation between increased blood loss during delivery and an increased risk of postpartum depression [27]. The authors suggested a dual etiology of developing the disorder—trauma experienced by the woman during hemorrhage and lowered iron levels affecting the mood. When studying the psychological sequelae of postpartum hemorrhage experienced by women, Sentihles et al., noted that the following factors were of importance—fear of death, fear for the child, pain experienced during the complication, and memories associated with traumatic delivery. Such experiences resulted in increased anxiety in the post-delivery period, difficulties with the relationships with a partner, disorders of sexual life, and fear of another pregnancy and delivery [28]. Psychological counselling was recommended to 60% of women in the study, but only 6% of them developed depressive disorders [28]. A significant role of hemoglobin deficiency as a factor predisposing to the development of postpartum mood disorders was emphasized by Corwin et al. They stated that postpartum anemia might be associated with the risk of developing postpartum depression [29]. Similar conclusions were presented by Etebary et al., who claimed that the reduced levels of iron, zinc, and magnesium might not be the only factors triggering postpartum depression, but they were important etiological factors of the disorder. The authors suggested the supplementation with multivitamin preparations as a method of supporting the therapy of postpartum depression. In particular, they recommended this alternative form of treatment during lactation in patients with the diagnosis of depression, who refused to use the possible conventional pharmacotherapy regimens, for fear of their unfavorable effect on the infant, and the possibility of becoming addicted to psychotropic agents [30]. The analysis of results obtained from a multicenter study supervised by Patricia Eckerdal, including women from Sweden, Finland, and Greece did not confirm the immediate correlation between the amount of blood lost during delivery and postpartum depressive disorders [31].

A neonatal disease was found to be a risk factor of postpartum depression in a study conducted in Ethiopia [32]. A study carried out in 4270 mothers whose children were placed in care at birth showed that the participants were characterized by a higher level of anxiety and postpartum depression (AOR = 2.31; 95% CI 1.08–1.59), compared to the control group (AOR = 2.13; 95% CI 1.67–2.73) [33]. Mothers who are involved with child protective services (CPS) usually have a lower financial status, have mental disorders, or are addicted to psychoactive substances, which are the reasons for placing children in care. Lower birth weight and lower Apgar score in the child appeared to be the risk factors of postpartum mood disorders. A lower Apgar score indicates less or more severe disorders with regards to certain parameters of the neonatal status (muscle tone, cardiac function (pulse), reflexes (reaction to stimuli), the color of the skin, and ventilation). A lower birth weight might herald an intrauterine developmental disruption, the presence of congenital defects, diseases, or other abnormalities. The suspicion of any developmental disorder in the child is the source of enormous stress for the mother and a risk factor of postpartum depressive disorders.

A study conducted in Tehran revealed that delivering a girl was a risk factor of developing postpartum depressive disorders, compared to delivering a boy (69% vs. 46.3%, respectively) [34]. The present study showed no correlation between the gender of the offspring and the risk of developing postpartum depressive disorders. A study conducted in Nepal by Maharjan et al., revealed the percentage of postpartum depression of 15.2% with the risk factor of postpartum depression being the gender of the offspring, especially in terms of the family’s preferences in the matter. Other risk factors included neonatal health problems and the presence of a partner during delivery [35].

It is believed that maternal physical and mental attachment is shaped throughout pregnancy and might contribute to reduced fetal growth. Mothers diagnosed with postpartum depression usually delivered children with lower birth weight, which was also confirmed in the present study [36,37]. A similar study conducted in Japan revealed no differences with regards to the intensification of postpartum depression, depending on the form of feeding the child [38].

The majority of available studies concentrated on postpartum depression. Therefore, some risk factors might not influence the occurrence of the common phenomenon of baby blues. Retrospective research is planned in this area.

The results of studies conducted so far are still ambiguous. Notably, the above-mentioned studies were conducted over a variety of time perspectives and socio-cultural settings. The views and attitudes towards motherhood, the expectations of women, and the expectations that other people have of women, change in societies. Therefore, research should be conducted in order to identify factors that might be present in prophylactic programs, with respect to postpartum mood disorders. A study conducted in Las Vegas in women who delivered in a private facility and were offered prenatal care and women who were not provided with prenatal care, revealed a twofold increase in the risk of postpartum depression (10.9% vs. 21.1%) [39]. It confirmed the assumed significance of the lack of support (also instrumental) in the etiology of postpartum depression.

### 4.1. Strengths of the Study

The authors of the present study would like to emphasize that the strengths of the present study include the analysis of a high number of case histories, which facilitated data collection with regards to specifying the study and control groups, and paying attention to the environmental factors that were not commonly analyzed in terms of postpartum mood worsening in women, in the early postpartum period.

The number of participants with high EPDS score was low (12% of all participants), which was the advantage of a comparative study of two extreme groups with relation to a continuous variable. It would not be possible to present the characteristics of such a small subgroup in such a large group of participants.

### 4.2. Limitations of the Study

This was a retrospective study, so it did not fully comprise some perinatal factors and circumstances associated with delivery, which might affect the emotional status of women after delivery. The prevalence of PPD also depends on the cut-off value of the study. A Portuguese study, in which an analogous EPDS was used, revealed that the percentage of reported PPD cases depended on the cut-off value and was 27.5% for scores >9 points, and 14.2% for scores >12 points [40]. It was also difficult to state explicitly that the identified perinatal predictors of mood worsening in women after delivery were correlated with it as such, as there was no information concerning intermediate factors that might enhance the results and conclusions of the analyses.

Regrettably, the time of hospitalization was a variable analyzed after measuring EPDS. Therefore, it could not be determined as a risk factor.

The present study focused on the peripartum factors. The psychosocial factors could not be determined, as it was a retrospective study. During a typical medical history, no questions concerning psychosocial support are asked.

The study also did not include a pre-delivery evaluation of depression status or risk factors among the two groups, which might constitute another limitation of the study.

## 5. Conclusions

The obtained results and conclusions might present a practical value for persons providing perinatal care at every stage, with particular attention paid to delivery as a culmination of pregnancy both for women and for the medical personnel.

A significant role in the estimation of the risk of developing postpartum mood disorders is played by rapid diagnosis, aiming at distinguishing a group of an increased risk of postpartum depression and the rapid implementation of preventive measures, both in the form of support and psychological and psychiatric treatment. Based on the obtained results, we concluded that increased supervision and support should be offered to women who experienced perinatal hemorrhage, who underwent instrumental deliveries, whose offspring were in a poorer physical condition, and who were hospitalized for longer periods.

The obtained results suggest the necessity of performing screening tests by medical personnel, taking care of women immediately after labor, and not only during the so-called postnatal check performed by midwives. Such tests would facilitate early identification of the risk of developing fully-symptomatic depression. Women whose offspring are in a poorer physical condition after delivery, and those who underwent instrumental delivery or developed perinatal hemorrhage, should be paid particular attention by the personnel.

Moreover, care provided by midwives should be based on the bidirectional monitoring and observation of women. First, to observe the physical status and condition of women who underwent a difficult delivery, and second, the identification of support which might be offered to the woman in labor by her relatives, and the assessment of the woman’s competences in providing care to the neonate, with particular attention paid to feeding. This would allow the identification of difficulties and strengthening the competences of women as mothers, and then, in adjusting the requirements of neonatal care to the condition and the physical capacity of the women. It might reduce the occurrence of the feeling of being devoid of such competences. It is a significant risk factor of developing peripartum mood worsening, and, as a consequence, postpartum depression. Medical personnel might considerably limit or eliminate this factor in the early postpartum period.

## Figures and Tables

**Table 1 ijerph-17-08726-t001:** Demographic characteristics of study participants.

Variable	N	%
**Education**		
primary or vocational	7	4.70%
secondary	43	28.90%
tertiary	100	66.40%
**Marital Status**		
married	124	0.83
unmarried	23	0.15
divorced	3	0.02
**Place of Residence**		
city	82	54.70%
town	38	25.30%
village	30	0.2

**Table 2 ijerph-17-08726-t002:** The comparison of variables associated with the course of delivery in the participants (the Mann–Whitney U test).

	EPDS	N	Mean	Standard Deviation	Z	*p*
Duration of stage I of delivery (minutes)	high	38	289.26	144.649	−0.602	0.547
	low	65	299.42	127.624		
Duration of stage II of delivery (minutes)	high	38	31.29	31.867	−1.791	0.07
	low	65	22.26	24.324		
Duration of stage III of delivery (minutes)	high	37	8.00	5.416	−1.947	0.05
	low	65	6.54	3.446		
Blood loss (mL)	high	64	410.94	180.050	−4.077	0.0001
	low	74	294.59	93.475		
Duration of hospitalization (days)	high	75	6.69	5.162	−3.098	0.002
	low	75	4.93	3.790		

**Table 3 ijerph-17-08726-t003:** Modes of delivery in the groups (chi-square test).

			EPDS		Chi-Square	*p*
			High	Low		
Mode of delivery	Vaginal delivery	N	40	66	21.741	0.0001
		%	53.3%	88.0%		
	Cesarean section	N	35	9		
		%	46.7%	12.0%		

**Table 4 ijerph-17-08726-t004:** Comparison of episiotomy rates in the groups (chi-square test).

			EPDS		Chi-Square	*p*
			High	Low		
Episiotomy	0	N	40	18	13.606	0.0001
		%	53.3%	24.0%		
	1	N	35	57		
		%	46.7%	76.0%		

**Table 5 ijerph-17-08726-t005:** Comparison of the frequency of the instrumental revision of the uterine cavity in the groups (chi-square test).

			EPDS		Chi-Square	*p*
			High	Low		
Revision	0	N	18	35	4.099	0.04
		%	62.1%	83.3%		
	1	N	11	7		
		%	37.9%	16.7%		

**Table 6 ijerph-17-08726-t006:** The comparison of variables associated with the neonatal status in the groups (the Mann–Whitney U test).

	EPDS	N	Mean	Standard Deviation	Z	*p*
APGAR	high	73	9.16	1.528	−2.048	0.04
	low	74	9.55	1.326		
Neonatal birth weight	high	73	3121.37	753.328	−2.672	0.008
	low	75	3469.47	497.366		

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
