# Peer review of "Peripartum Predictors of the Risk of Postpartum Depressive Disorder: Results of a Case-Control Study"

_ijerph, 2020, doi:10.3390/ijerph17238726_

Round 1
Reviewer 1 Report
To determine risk factors for postpartum depression (PPD), focusing on birth issues, the authors compared women who had been hospitalized for their delivery based on medical records, 75 with high scores on the EPDS and 75 with low scores. Those in the high-risk group were hospitalized significantly longer, had greater blood loss, were in labor significantly longer, were more likely to have a C-section delivery, and had a low birth weight infant. It was concluded that greater supervision and support should be offered to women with these risk factors, to prevent PPD.
The study findings appear to be original and are of great interest. The methods were appropriate for the study and the conclusions are sound. A major strength of the study is that it included a large number of cases, which provided data on environmental factors that are not usually analyzed in terms of postpartum mood worsening in the early postpartum period. Limitations included the fact that the study was retrospective and data pertaining to perinatal factors and circumstances associated with delivery may have been overlooked. The study was also not driven by hypothesis and the authors have not speculated about how the factors they identified relate to the causes of PPD. My only recommendation is that they might consider adding a few lines on the pathogenesis of PPD.
Author Response
Dear Reviewer,
At the beginning, we want to thank you for all valuable remarks. We are happy that you enjoyed this topic. The manuscript has been revised according to your suggestions. Detailed answer is provided below.
Comments and Suggestions for Authors
My only recommendation is that they might consider adding a few lines on the pathogenesis of PPD.
It has been added in the Introduction.
Best regards
Kornelia Zareba and Co-authors
Reviewer 2 Report
1) Recognized risk factors for depression are also:
- Previous history of depression and anxiety
- Young age during pregnancy
- Social support: emotional support, financial support, intelligence support, and empathy relations (eg Ghaedrahmati et al. 2017)
The Authors reported some of these risk factors in the introduction, but they should include and discuss these factors in their analysis.
2) Based on their findings, what preventive measures do the Authors suggest?
Author Response
Dear Reviewer,
At the beginning, we want to thank you for all valuable remarks. We are happy that you enjoyed this topic. The manuscript has been revised according to your suggestions. Detailed answer is provided below.
Comments and Suggestions for Authors
- Recognized risk factors for depression are also:
- Previous history of depression and anxiety
- Young age during pregnancy
- Social support: emotional support, financial support, intelligence support, and empathy relations (eg Ghaedrahmati et al. 2017)
The Authors reported some of these risk factors in the introduction, but they should include and discuss these factors in their analysis.
It has been added in the Introduction and Discussion. The present study focused on peripartum factors. They were comprised in the following paper: Jolanta Banasiewicz, Kornelia ZarÄ™ba, MaÅ‚gorzata BiÅ„kowska, Hanna Rozenek, StanisÅ‚aw Wójtowicz and Grzegorz Jakiel. Perinatal Predictors of Postpartum Depression: Results of a Retrospective Comparative Study. J Clin Med. 2020, 9(9): 2952. The psychosocial factors could not be determined, as it was a retrospective study. During a typical medical history no questions concerning psychosocial support are asked (it has been added in the article).
- Based on their findings, what preventive measures do the Authors suggest?
It has been added in the Conclusions.
Best regards
Kornelia Zareba and Co-authors
Reviewer 3 Report
This is a hospital-based case-control study of risk factors of postpartum depressive disorder; cases were 75 women with a EPDS score ≥12 and controls were those with ≤5 points upon admission. The study found that women had a higher risk of development postpartum depressive disorder (using EDPS score as a proxy) were those who had more blood loss during delivery, longer labor stage, cesarean section, received curettage of the uterine cavity, etc. I have a few comments:
- The study design of “retrospective comparative study” is not very clear to me; is it similar to a case-control study?
- When was the EPDS evaluated? Was it right after delivery? Length of hospitalization seems to happen after EPDS and therefore does not seem to be a risk factor.
- How did you select the 75 cases and 75 controls, in addition to EPDS score?
- Since you mentioned EPDS score does not already predict postpartum depressive disorder, what are the advantages of the current design vs. analyzing all 604 patients with EPDS as a continuous variable?
- How comparable are the two groups, in addition to those risk factors studied? For example, was age distribution similar between the two groups?
- Table 1: suggest showing age in the table, as it’s important. It would also be clearer to show the variables by group.
- Table 2: Please add units for all variables when applicable. Also, are there any clinical significance for those differences between the two groups? Could you contextualize the results a bit more?
- Table 4: Is “4.7%” a typo?
- Table 6: Are there any clinical significance in the differences you observed in APGAR and birth weight? For example, if two newborns are of both of normal weight, and one was slightly heavier or lighter than another one, this does not seem to be a risk factor for depression for the mother?
- The study did not seem to have a pre-delivery evaluation of depression status or risk factors among the two groups, which seems to be a limitation worth mentioning.
- Have you done any multivariable analysis?
Author Response
Dear Reviewer,
At the beginning, we want to thank you for all valuable remarks. We are happy that you enjoyed this topic. The manuscript has been revised according to your suggestions. Detailed answer is provided below.
Comments and Suggestions for Authors
- The study design of “retrospective comparative study” is not very clear to me; is it similar to a case-control study?
Yes. We have added “Results of a Case-Control Study” to the title.
- When was the EPDS evaluated? Was it right after delivery? Length of hospitalization seems to happen after EPDS and therefore does not seem to be a risk factor.
Yes, the time of hospitalization is a variable measured after measuring EPDS. EPDS was measured 2-4 days after delivery. Indeed, it may not constitute its risk factor. It has been added to the text.
- How did you select the 75 cases and 75 controls, in addition to EPDS score?
Apart from the score we analyzed the age, marital status and the place of residence. It has been added to the text.
- 4. Since you mentioned EPDS score does not already predict postpartum depressive disorder, what are the advantages of the current design vs. analyzing all 604 patients with EPDS as a continuous variable?
The number of participants with high EPDS score is low (12% of all participants) which is the advantage of a comparative study of two extreme groups in relation to a continuous variable. It would not be possible to present the characteristics of such a small subgroup in such a large group of participants. It has been added to the Discussion section.
- How comparable are the two groups, in addition to those risk factors studied? For example, was age distribution similar between the two groups?
Age was one of the selection criteria, so the answer is obviously “yes”.
- Table 1: suggest showing age in the table, as it’s important. It would also be clearer to show the variables by group.
Age was a selection criterion. Therefore, only mean values were presented. The information has been added to the text.
- Table 2: Please add units for all variables when applicable. Also, are there any clinical significance for those differences between the two groups? Could you contextualize the results a bit more?
Units have been added throughout the text. The description of the results has been added to the text.
- Table 4: Is “4.7%” a typo?
Yes. It has been corrected.
- 9. Table 6: Are there any clinical significance in the differences you observed in APGAR and birth weight? For example, if two newborns are of both of normal weight, and one was slightly heavier or lighter than another one, this does not seem to be a risk factor for depression for the mother?
It was not assessed in the study.
- The study did not seem to have a pre-delivery evaluation of depression status or risk factors among the two groups, which seems to be a limitation worth mentioning.
The evaluation was carried out with stepwise logistic regression. As regards the analyzed variables, neonatal weight was the only significant (although not very strong) factor which underlay high or low EPDS score. The information has been added to the text.
Best regards
Kornelia Zareba and Co-authors
Round 2
Reviewer 2 Report
The manuscript is suitable for publication in its current form.
Reviewer 3 Report
Thank you for the detailed responses, I don't have further comments.